# Relatively Low Lecithin Inclusion Improved Gelling Characteristics and Oxidative Stability of Single-Washed Mackerel (*Auxis thazard*) Surimi

**DOI:** 10.3390/foods13040546

**Published:** 2024-02-10

**Authors:** Worawan Panpipat, Thinnaphop Chumin, Porntip Thongkam, Pattaraporn Pinthong, Kalidas Shetty, Manat Chaijan

**Affiliations:** 1Food Technology and Innovation Research Center of Excellence, School of Agricultural Technology and Food Industry, Walailak University, Thasala, Nakhon Si Thammarat 80160, Thailand; pworawan@wu.ac.th (W.P.); thinnaphop.ch@wu.ac.th (T.C.); pohntip23@gmail.com (P.T.); pattar1001@gmail.com (P.P.); 2Global Institute of Food Security and International Agriculture (GIFSIA), North Dakota State University, 374 D Loftsgard Hall, 1360 Albrecht Blvd., Fargo, ND 58108, USA; kalidas.shetty@ndsu.edu

**Keywords:** surimi, mackerel, lecithin, gel, lipid oxidation

## Abstract

The effect of lecithin addition on the gelling characteristics and oxidative stability of single-washed mackerel (*Auxis thazard*) surimi was investigated in this study. Surimi was chopped in the presence of 2.5% (*w*/*w*) NaCl with different concentrations of lecithin (0, 0.1, 0.5, 1, and 1.5 g/100 g surimi). The rheological behavior, gel-forming ability, microstructure, and lipid oxidation of lecithin-added surimi varied significantly depending on lecithin content. When compared to the control, lecithin at 0.1, 0.5, and 1 g/100 g improved the breaking force of the gel (*p* < 0.05). The breaking force of the gel decreased significantly as lecithin concentration increased (up to 1.5 g/100 g) (*p* < 0.05). Deformation, on the other hand, reacted differently to the lecithin than it did to the breaking force. At a lecithin level of 0.1 g/100 g, the surimi gel displayed improved deformation (*p* < 0.05). Nonetheless, at higher doses (0.5–1.5 g/100 g), lecithin considerably reduced surimi gel deformation (*p* < 0.05), and the gel containing lecithin at 1.5 g/100 g showed significantly decreased deformation. Surimi with 0.1 g/100 g lecithin had the lowest expressible drip (*p* < 0.05). In general, lecithin at concentrations ranging from 0.1 to 1 g/100 g reduced expressible drip (*p* < 0.05), but not at 1.5 g/100 g, which was equivalent to the control (*p* > 0.05). Adding lecithin to mackerel surimi improved its whiteness slightly, regardless of concentration. Lecithin impacted the microstructures of surimi gel in a concentration-dependent manner. Lecithin at a concentration of 0.1 g/100 g produced a densely packed network with small, jointed clusters and minimal holes within the gel. Joined clusters in the gel were reduced by 0.5–1.5 g/100 g lecithin, and continuous aggregates predominated. Surprisingly, at higher doses of lecithin, notably 1.5 g/100 g, porous structures with continuous voids were perceived. Surimi gels treated with various lecithin doses had lower thiobarbituric acid reactive substances (TBARS) levels than the control (*p* < 0.05). Overall, lecithin at a low concentration of 0.1 g/100 g was most effective at improving the texture, increasing water-holding capacity, lightening the color, and delaying lipid oxidation of single-washed mackerel surimi.

## 1. Introduction

Frigate mackerel (*Auxis thazard*) is a typical dark-fleshed fish caught in Thailand. In total, around 3.7 metric tons of this species were caught in Thailand in 2022 from the Indian Ocean and the Gulf of Thailand [1]. This species is currently a feasible source to produce surimi, and the optimal washing technique to maximize the sustainability of this species has been systematically established [2,3,4]. Washing mackerel (*A. thazard*) minced twice in cold water following a cycle of cold carbonated water with 0.6% (*w*/*v*) NaCl (soda-saline solution; SSS) is a simple approach for increasing the gel-forming capacity and oxidative stability of the finished surimi [2]. To make it more sustainable and eco-efficient, a feasible single washing process for producing mackerel (*A. thazard*) surimi was established using ultrasonication coupled with SSS and mixed antioxidants (0.5% EDTA + 0.2% sodium tripolyphosphate + 0.2% sodium erythorbate; referred to as antioxidant infused SSS) [3,4]. This could advance the manufacturing of dark-fleshed fish surimi, which would require less time to wash and generate less wastewater [4]. This method, also referred to as the lipid-stabilizing strategy, was developed to limit oxidation and rancidity, but the potential of the resulting surimi to form an ideal gel remained a difficulty. Thus, utilizing functional additives to improve the gel characteristics of single-washed mackerel surimi could be an attractive route forward.

Lecithin is a phospholipid combination containing phosphatidylcholine, phosphatidylserine, and phosphatidylinositol obtained from both animal and vegetable origins [5]. Phosphatidylcholine, a major constituent in lecithin, is a phospho-*O*-glyceride containing choline that is ester-bonded to phosphoric acid [5]. The first and second positions of glycerol include varying quantities of residues from different unsaturated and/or saturated fatty acids. They are linked to glycerol via ester linkages. Saturated fatty acids are typically found in the first position. Phosphorylcholine is found in the third position of glycerol [6]. Phosphatidylcholine has an average molecular weight of 768 g/mol [5]. The pH of the phosphatidylcholine solution varies between 6.6 and 7 [7]. The US Food and Drug Administration (FDA) classifies phosphatidylcholine as GRAS (Generally Recognized as Safe). Even at large doses, no significant side effects have been reported [5].

Lecithin’s market has grown because of the growing interest in plant-based foods, its numerous industrial uses, and its vital function in health advantages [8]. Lecithin is used not just in foods but also in cosmetics, lubricants, and feeds [9,10,11]. Lecithin is the most significant natural emulsifier in the food sector. Manufacturers frequently anticipate benefitting from lecithin’s antioxidant activity, in addition to its usage as a food emulsifier. Thus, lecithin is a key natural emulsifier with antioxidant activity that is frequently employed in food preparation [11]. Furthermore, lecithin is a lipid that is required for cell membranes and brain functioning. Lecithin has been shown to increase cognitive performance, boost immunological function, control lipid metabolism, lower cholesterol levels, and lower cardiovascular risk [12,13]. Soy lecithin supplementation has previously been shown to alter the activities of macrophages and lymphocytes, implying the immunomodulatory effects of soy lecithin [14].

It is paradoxical that adding a phospholipid either makes the surimi gel stronger or weaker. The composition and content of phospholipids in surimi affected its textural qualities and gel strength [15]. It has been documented that the surimi made from bighead carp (*Aristichthys nobilis*) that still contains phosphatidylserine and sphingomyelin improves the surimi’s chewiness, hardness, and gel strength. More disulfide bonds and hydrophobic contacts were shown to improve the gel strength in bighead carp surimi by stabilizing the gel network structure [15]. The inclusion of large yellow croaker (*Larimichthys crocea*) roe phospholipids could improve the gel strength, textural qualities, rheological features, and water-holding capacity (WHC) of sea bass (*Lateolabrax japonicas*) surimi gels [16]. Lecithin’s amphiphilic nature with a positive choline group and a negative phosphate moiety potentially benefited the establishment of an equilibrium of protein–protein bonding and protein–water association in gel networks for application in food protein gels [17,18]. In the study by Xia et al. [19], lecithin’s polar head may generate hydrogen bonds with water upon heat gelation, while its non-polar tail may make hydrophobic contact with the hydrophilic part of pork myofibrillar proteins. Zhou et al. [12] proposed using lecithin at 0.4–0.8 g/100 g for the manufacture of healthier gurnard (*Lepidotrigla microptera*) surimi. Lecithin could be utilized to enrich reduced-salt surimi gels prepared from Amur sturgeon (*Acipenser schrenckii*). When lecithin was introduced at 0.1 and 0.3 g/100 g, there was an upward trend in hydrogen bonding and β-sheet production [20]. According to Xia et al. [19], the addition of lecithin changed the textural characteristics, WHC, and whiteness of the gel. This evolution could be attributed to an increase in hydrogen bonds and hydrophobic interactions. Recently, lecithin at 1 g/100 g was shown to be the ideal concentration for the overall quality of surimi gel made from bigeye snapper (*Priacanthus tayenus*) [18]. According to those reports, the impact of lecithin on surimi gel properties was determined by the type of fish, the grade of surimi raw material, the concentration of lecithin, and the method of lecithin preparation.

The impact of lecithin on the gelling capabilities of mackerel surimi, on the other hand, has not been investigated. Thus, the purpose of this study was to determine the effect of lecithin incorporation at various concentrations on the gel properties of single-washed frigate mackerel (*A. thazard*) surimi. The results collected can be used to advance health-oriented surimi products prepared from dark-fleshed fish, adhering to the growing demand for products with functional ingredients.

## 2. Materials and Methods

### 2.1. Chemical and Materials

The food-grade soy lecithin was purchased from Chemipan Corporation Co., Ltd. in Bangkok, Thailand. All chemicals used in the analysis such as 1,1,3,3-tetramethoxypropane (TMP), trichloroacetic acid (TCA), and thiobarbituric acid (TBA) were of analytical grade and ordered from Sigma Aldrich Co. (St. Louis, MO, USA).

Mackerel (*A. thazard*) measuring 100–120 g/fish were bought in Thasala, Nakhon Si Thammarat, Thailand, from a local market. The fish were collected roughly 12 h after capture and encased in ice with a fish/ice weight proportion of 1:3 (*w*/*w*) before being delivered to Walailak University within 20 min. The fish were then headed, eviscerated, washed, filleted, and skinned. Following that, the fish flesh was completely chopped with a meat grinder with a 4 mm aperture pitch (Panasonic MK-G20MR, Tokyo, Japan).

### 2.2. Surimi Preparation

Surimi was produced using a single ultrasound-assisted washing cycle of 3 vol of cold carbonated water in the presence of 0.6% NaCl and a mixture of antioxidants (0.2% sodium tripolyphosphate/0.2% sodium erythorbate/0.5% EDTA) for 5 min [4]. An ultrasonicator (Sonics, Model VC750, Sonica & Materials, Inc., Newtown, CT, USA) was used to carry out an ultrasound treatment. A 25 mm diameter flat-tip probe was employed, with an amplitude of 80% and an intensity of 153 W/cm^2^ at a single frequency of 20 kHz. Following washing, the minced fish was run through a layer of nylon screen to remove water, and then it was hydraulically compressed until it had a final moisture content of roughly 78–80%. Before freezing in a freezer (Polar DN494 367 Blast Freezer, Campbell Town, NSW, Australia), washed mince was well blended with 4% (*w*/*w*) sorbitol and 4% (*w*/*w*) sucrose. The frozen surimi was then kept at −18 °C until it was examined within a month.

### 2.3. Effect of Lecithin on the Rheological Properties and Characteristics of Surimi Gel

Pieces of surimi were chopped for a five-minute period with 2.5% (*w*/*w*) NaCl and varying amounts of lecithin (0, 0.1, 0.5, 1, and 1.5 g/100 g of surimi) by a Panasonic food processor (MK-K51PKSN, Selangor Darul Ehsan, Malaysia). The lecithin concentration employed in this study was based on our prior evidence, which suggested using lecithin at 1 g/100 g for bigeye snapper surimi [18]. The type of surimi raw material and the washing procedure, however, may play a vital role in determining how much lecithin is integrated. As a result, for the single-washed mackerel surimi, lecithin concentrations at lower and higher than 1 g/100 g were used in this investigation. Before applying lecithin, it was dissolved in a small amount of chilled water. The final moisture content in the surimi mixture for all treatments was adjusted to 80% (*w*/*w*). The resultant homogeneous sol was separated into two groups after chopping. The first group was for rheological analysis, whereas the second was for gel preparation.

A rheometer with a 35 mm diameter parallel plate was employed for the rheological analysis (HAAKE MARS 60; Thermo Fisher Scientific Inc., Yokohama, Japan). With a constant frequency of 1 Hz and an amplitude strain of 2%, gelation was started by sweeping the temperature from 20 °C to 90 °C at an average pace of 2 °C/min. The elastic modulus (G′), viscous modulus (G″), and loss tangent (tan δ) data were provided [18].

To prepare the gel, the sol was placed inside a polyvinylidine shell with a 2.5 cm diameter, and rubber bands were used to seal the ends. Next, cooking was performed for 20 min at 90 °C in a controlled temperature water bath (W350 Memmert, Schwabach, Germany) after it had been set for 30 min at 40 °C. [18]. The gel samples were submerged for 30 min in ice water, kept at 4 °C for 24 h, and then used for evaluation.

### 2.4. Analysis of Gelling Properties

Using the technique of Buamard and Benjakul [21], gel properties such as breaking force, deformation, whiteness, and expressible drip were examined. Breaking force and the deformation of gel with a 2.5 cm length were determined using a TA-XT2i texture analyzer (Stable Micro Systems, Godalming, Surrey, UK) equipped with a spherical plunger (5 mm diameter and speed of 60 mm/min). The expressible drip was calculated based on the percentage of the sample weight before and after being compressed by a 5 kg standard weight for 2 min where a thinly sliced sample (5 mm thickness) was placed between pieces of Whatman No. 1 filter paper. The color parameters, including *L**, *a**, and *b** values, were recorded using a portable Hunterlab Miniscan/EX instrument (Hunter Assoc. Laboratory, Reston, VA, USA), and the whiteness was calculated by the following formula:Whiteness = 100 − [(100 − *L**)^2^ + *a**^2^ + *b**^2^]^1/2^(1)

### 2.5. Analysis of Microstructure

Surimi gel microstructure examination was carried out using the scanning electron microscope (SEM) (GeminiSEM, Carl Ziess Microscopy, Oberkochen, Germany) [2]. Samples with a thickness of 2–3 mm were fixed with 2.5% (*v*/*v*) glutaraldehyde in 0.2 M phosphate buffer (pH 7.2) for 2 h. The samples were then rinsed for 1 h in distilled water before being dehydrated in ethanol with serial concentrations of 50, 70, 80, 90, and 100% (*v*/*v*). Dried samples were mounted on a bronze stub and sputter coated with gold. The specimens were observed with an SEM at an acceleration voltage of 5 kV.

### 2.6. Analysis of Lipid Oxidation

As a measure of lipid oxidation, thiobarbituric acid reactive substances (TBARS) of surimi gels were examined [22]. A ground sample (0.5 g) was homogenized with an IKA^®^ homogenizer (Model T25 digital Ultra-Turrax^®^, Staufen, Germany) with 2.5 mL of a solution containing 0.375% (*w*/*v*) TBA, 15% (*w*/*v*) TCA, and 0.25 M HCl at 9500 rpm for 2 min in an ice bath. The mixture was heated in a boiling water bath (95–100 °C) for 10 min, cooled with running tap water, and then centrifuged (3600× *g*/25 °C/20 min). The absorbance of the supernatant was measured at 532 nm. A standard curve was prepared using 1,1,3,3-tetramethoxypropane at concentrations ranging from 0 to 10 mg/L. Malondialdehyde (MDA) equivalent in mg/kg of gel was reported as the TBARS content.

### 2.7. Statistical Analysis

A completely randomized design (CRD) was applied for all examinations. Data are presented as mean ± standard deviation from triplications. The Statistical Package for the Social Sciences for Windows (SPSS Inc., Chicago, IL, USA) was used to process the data. Significant differences (*p* < 0.05) between samples were found using Duncan’s multiple-range analysis.

## 3. Results and Discussion

### 3.1. Rheology

The dynamic thermal gelation patterns of surimi added with varied lecithin concentrations from 20 to 90 °C regarding G′, G″, and tan δ are shown in Figure 1. G′ describes a sample’s elastic behavior in general by monitoring the deformation energy stored in it during shearing [3,23]. Denaturation and the association of several myofibrillar proteins at various temperatures induced G′ alterations during heating. According to the results, G′ increased and peaked around 40–45 °C for all samples (Figure 1a). A rise in G′ indicated that the firmness of the sample had increased due to the establishment of an elastic structure [23]. The first protein network was created during the “gel setting” process. At this time, myosin would unfold to allow for organized polymerization, and the early elasticity of proteins would have dissipated [24]. This range covers the creation of protein networks via weak interactions between protein molecules, such as hydrogen bonds, according to Buamard et al. [25]. Then, G′ decreased immediately, touching its lowest point at 50 °C for surimi with no or low lecithin content (0–0.5 g/100 g) and about 60 °C for surimi with 1–1.5 g/100 g, suggesting gel weakening. It was most likely due to residual protease activity at this temperature, which can speed up myosin degradation and interfere with gel formation [26]. Higher concentrations of lecithin (1 g/100 g) may modify the G′ pattern, most likely due to lecithin’s propensity to increase the heat stability of endogenous protease. Some heat-activated fish proteases are activated between 50 and 60 °C, which is close to their optimum temperature [26]. Surimi was set at 40 °C and cooked at 90 °C to avoid gel weakening caused by remaining proteases.

G’ was subsequently increased again, peaking around 70 °C for surimi combined with lecithin at 0.1 g/100 g and 84–85 °C for other samples, indicating the formation of a robust gel network due to higher attractive forces (e.g., disulfide bond and hydrophobic interactions). Gelled surimi with lecithin at 0.1 g/100 g appeared to generate a stronger gel network earlier than other samples. The predominant cross-linking and aggregation of myosin occurred during this “gel strengthening” stage. While oxidizing sulfhydryl (SH) groups, a disulfide bridge may have formed, and hydrophobic domains may have been connected via hydrophobic–hydrophobic linkage [23]. G′ was then further lowered at high temperatures. One possible explanation for this could be that hydrogen bonds disintegrate at high temperatures [23].

Figure 1b depicts the surimi viscosity modulus (G″) curves. G″ is typically used to measure the viscosity of the surimi gel and refers to the loss modulus [23]. G″ exhibited a similar pattern as G′, though the extent of organized interaction was considerably more powerful, leading to a lesser G″ value, suggesting that the elastic structure was the most crucial element in the production of surimi gel [4]. G″ increased until it reached about 45 °C, and then decreased swiftly to 50 °C for surimi with no or low levels (0–0.5 g/100 g) of lecithin. Additionally, surimi with low or no lecithin levels (0–0.5 g/100 g) had nearly identical G″ values up to 65 °C. However, two peaks between 45 and 55 °C were seen in samples with high amounts of lecithin (1–1.5 g/100 g), indicating a structural shift at these temperatures. At 90 °C, which was used for cooking during thermal-induced gelation, the G″ of surimi with no or low levels of lecithin (0–0.5 g/100 g) was higher than those with high levels of lecithin (1–1.5 g/100 g). Thus, it appears that lecithin had a concentration-dependent effect on the viscoelastic characteristics of surimi, which was linked to the gel strength of surimi.

Tan δ for all samples was less than 1.0 because G′ values were greater than G″ values (Figure 1c). As a result, every sample displayed the properties of an elastic fluid with enhanced gelling ability [27]. Tan δ differed between two sets of lecithin levels that followed the pattern of G′ and G″, showing that all surimi had undergone varying degrees of sol–gel transition after heating. Consequently, due to the rheological behavior, two-step heating (40 °C/30 min and 90 °C/20 min) can be used for the production of gels where all samples can gel to varying degrees.

### 3.2. Gelling Properties

#### 3.2.1. Breaking Force and Deformation

The effects of lecithin addition at different levels on the breaking force and deformation of surimi gels are presented in Figure 2a and 2b, respectively. When heated, the myofibrillar proteins in salted surimi pastes reveal reactive surfaces, which collaborate to create intermolecular interactions [20]. A three-dimensional network arises when there are enough intermolecular bonds, producing a surimi gel. Hydrogen bonds strengthen the hydrogel during the cooling phase and help to stabilize the bound water inside of it. Temperature-induced strengthening of intermolecular hydrophobic contacts is thought to be the main mechanism underlying the creation of surimi gel [20].

For the breaking force (Figure 2a), lecithin at 0.1, 0.5, and 1 g/100 g showed an improved breaking force, a rough increase of 4–6%, when compared to the control (without lecithin addition) (*p* < 0.05). When raising the lecithin level (up to 1.5 g/100 g), the breaking force of the gel dropped to about a 7% reduction (*p* < 0.05). The ideal level of lecithin to integrate into surimi has been reported to be species dependent. To produce surimi gel from Amur sturgeon (*Acipenser schrenckii*) surimi [20] and gurnard (*Lepidotrigla microptera*) surimi [12], respectively, lecithin at 0.1 g/100 g and 0.4–0.8 g/100 g was recommended. The optimal amount for lecithin was found at 1 g/100 g to produce bigeye snapper surimi gel [18]. The gel’s breaking force decreased as the lecithin level was raised above this concentration [12,18,20].

Deformation, however, responded to the lecithin in a different way than it did to the breaking force (Figure 2b). The surimi gel exhibited enhanced deformation (~3%) at a lecithin level of 0.1 g/100 g, the highest value (11.56 mm) (*p* < 0.05). Nonetheless, lecithin significantly decreased the surimi gel’s deformation at higher concentrations (0.5–1.5 g/100 g) (*p* < 0.05), and the gel containing lecithin at 1.5 g/100 g showed significantly decreased deformation (~20% decrease).

It has been suggested that lecithin may bind water strongly at low concentrations without impairing gel formation. Nevertheless, lecithin could interact with water at high concentrations and impair the interaction between proteins. As a result, when a high content of lecithin was utilized, the breaking force and deformation greatly decreased. The results matched the impact of other conventional oils on the surimi gel’s textural characteristics. According to Benjakul et al. [28], the breaking force of mixed surimi gel, which contains both bigeye snapper and mackerel, reduced as the amount of soybean oil increased. As the virgin coconut oil (VCO) level increased, the breaking force and deformation of the croaker surimi gel reduced [29]. Consequently, to obtain the appropriate gel strength, attention should be paid to the lecithin content.

#### 3.2.2. Expressible Drip and Whiteness

Figure 3a depicts the impact of lecithin inclusion on the expressible drip of single-washed mackerel surimi gel. Gel with a lower expressible moisture exhibited a higher WHC in general [30]. Surimi with lecithin at 0.1 g/100 g showed the lowest expressible drip (*p* < 0.05). In general, lecithin at 0.1–1 g/100 g reduced expressible drip (*p* < 0.05), but not at 1.5 g/100 g, which was equal to the control (*p* > 0.05). Panpipat et al. [18] demonstrated that the expressible drip of threadfin bream surimi decreased with increasing lecithin content up to 1 g/100 g, but then increased at 2–3 g/100 g.

The finding was substantiated by breaking force and deformation results (Figure 2a,b), which showed that lecithin at 1.5 g/100 g had a negative influence on gel strength and deformability. Amphiphilic lecithin could bind with both hydrophilic and hydrophobic structures in the gel. As a result, lecithin at low concentrations (0.1 g/100 g) may bond thoroughly to water, preventing water release and improving WHC. However, at high concentrations (i.e., 1.5 g/100 g), lecithin may form a multilayer water framework that could be simply squeezed out, as some studies have proposed. The gel network is unable to physically entrap loosely bound layered water, causing water release [18]. Water may be retained in the gel at a suitable lecithin content because lecithin can bind water through the polar portion [19]. Chemical as well as physical trapping can raise the gel’s WHC [18].

Figure 3b depicts the whiteness of gels made from single-washed mackerel surimi with varying lecithin concentrations. Whiteness is a key determinant of surimi gel quality [30]. Normally, the color of surimi gel was impacted by the particular type and level of food additives [28]. Vegetable oil, such as soybean oil, corn oil, and VCO, both in bulk and pre-emulsified forms (i.e., nanoemulsion), have been used to increase the surimi gel’s whiteness [29,31,32]. The reason for this is that the surimi gel’s increased light scattering makes it appear whiter [18,28]. In particular, the light scattering potential of the emulsion generated following chopping resulted in a lighter surimi gel when combined with oil [28]. According to Benjakul et al. [28], adding 1–5 g/100 g of soybean oil can enhance the whiteness of mixed surimi. In accordance with Pietrowski et al. [31], the corn oil-infused Alaska pollock surimi gel was likewise lighter than the control. The gel’s color became whiter than the control when surimi was mixed with VCO in the form of bulk oil and nanoemulsion [33]. Oil is mostly made up of triglycerides; however, it may have different gel strengthening and whitening properties from lecithin, a polar lipid [18]. It has been shown that lecithin enhanced the surimi gel’s whiteness by enhancing the way that light was reflected. Lecithin incorporation at 1–3 g/100 g previously increased the whiteness of threadfin bream surimi [18]. Adding lecithin to mackerel surimi increased its whiteness marginally in this study, regardless of concentration. This was because mackerel muscle had a darker color and was more difficult to whiten than fish with white flesh, such as threadfin bream. According to the proposed approach outlined by Panpipat et al. [18], lecithin may align at the interface to form an emulsion in the system, which can improve the whiteness of surimi gel. Therefore, because of the introduction of the emulsion, it might enhance the whiteness of surimi gel [33]. Lecithin and muscle proteins have an emulsifying action; therefore, an emulsion should occur to some extent during chopping in the presence of lecithin. This occurrence may have contributed to the whiteness of the resulting gel. Figure 4 depicts the appearance of surimi gel in this study.

### 3.3. Microstructure

SEM images revealed that the microstructure of the gel samples differed (Figure 5). An ideal surimi gel had finer and more ordered networks, whereas a poor gel had a discontinuous network with bigger pores [2,3,4]. Surimi gel without lecithin (Figure 5a) exhibited heterogeneous aggregates with occasional spaces, as well as minute particle clots and a continually tightly packed structure. The addition of lecithin altered the microstructures of surimi gel in a concentration-dependent manner (Figure 5b–e). Lecithin has been shown to change the secondary structures and hydrophobicity of proteins, which improves the gel’s properties by making the gel network denser and more homogeneous [20]. This was because lecithin included negatively charged phosphate groups and positively charged choline groups that could form ionic bonds with myofibrillar proteins. This altered the spatial conformation of the myofibrillar proteins and encouraged their binding and aggregation, which increased the gel network [16]. Lecithin levels of 0.1 g/100 g resulted in a tightly packed network with small, jointed clusters and some minor gaps in the gel. Jointed clusters were reduced in the gel with 0.5–1.5 g/100 g lecithin, and continuous aggregates predominated. Surprisingly, porous structures with discontinuous voids were observed at greater levels of lecithin, particularly at 1.5 g/100 g. It has been noted that high concentrations of lecithin (i.e., 2–3 g/100 g) may produce an imbalance in the two interlinks, protein–protein and protein–water. As a result, voids developed, and the breaking force and deformation rapidly decreased, while the expressible drip grew significantly [18]. This phenomenon was also observed in this investigation when 1.5 g/100 g lecithin was used. Zhou et al. [12] discovered that lecithin concentrations of more than 1.6 g/100 g had a deleterious impact on the structural feature of gurnard surimi gel.

### 3.4. TBARS

Phospholipids, particularly lecithins, are widely utilized as emulsifiers and are gaining popularity as natural antioxidants to limit lipid oxidation [11]. Numerous theories have been put forth regarding how phospholipids might affect lipid oxidation. Phospholipids have the ability to bind prooxidative metals, creating antioxidative molecules during lipid oxidation by means of Maillard reactions, changing the position of other antioxidants, and, furthermore, renewing primary antioxidants, like tocopherols. Phospholipids, on the other hand, may also act as their own oxidation substrates. Because of their high degree of unsaturation, there is significant surface area when they appear in the form of dispersions and allow the negative charge to bind prooxidant metals; therefore, they can become a key target for oxidation in foods that include significant amounts of biological membranes, like animal products [11].

The TBARS values of surimi gels supplemented with varying lecithin concentrations are displayed in Figure 6. When lecithin content increased to 1 g/100 g, the TBARS values of the lecithin-infused surimi gel dramatically decreased in comparison to the control group (*p* < 0.05). When lecithin was increased to 1.5 g/100 g, the TBARS value remained unchanged (*p* > 0.05). The outcome matched a prior study on bigeye snapper surimi, which found that lecithin at a concentration of 1 g/100 g was the best for maintaining the gel texture, enhancing the WHC, lightening the color, and delaying lipid oxidation [18].

## 4. Conclusions

This study investigated the influence of lecithin addition on the gelling qualities and oxidative stability of single-washed mackerel (*A. thazard*) surimi, with emphasis on the rheological behavior, gel-forming ability, microstructure, and lipid oxidation. All the quality criteria of lecithin-added surimi gel differed dramatically depending on the amount of lecithin present. Overall, a low level of lecithin (0.1 g/100 g) improved breaking force and deformation greatly, whereas a high level (1.5 g/100 g) had a negative impact on gel texture. Furthermore, surimi with 0.1 g/100 g lecithin had the lowest expressible drip. Regardless of concentration, adding lecithin to mackerel surimi boosted its whiteness slightly. Lecithin has a concentration-dependent effect on the microstructures of surimi gel. A concentration of 0.1 g/100 g lecithin created a densely packed network with small, jointed clusters and few holes inside the gel. Surimi gels treated with different amounts of lecithin had lower TBARS levels than the control. Overall, lecithin at 0.1 g/100 g was the most effective at improving texture, increasing WHC, whitening the color, and retarding lipid oxidation in single-washed mackerel surimi. However, sensory evaluation and storage stability can be further investigated to meet industrial requirements. In addition, the combined effect of lecithin and other functional food ingredients on the gel characteristics of mackerel surimi can be investigated in the future.

## Figures and Tables

**Figure 1 foods-13-00546-f001:**
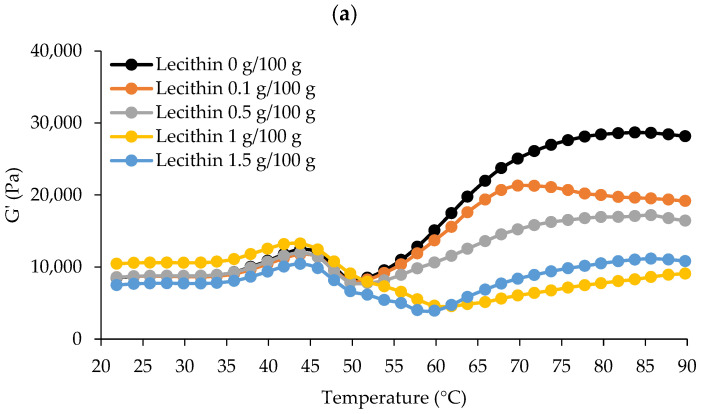
The dynamic thermal gelation patterns of single-washed mackerel surimi added with varied lecithin concentrations in terms of G′ (**a**), G″ (**b**), and tan δ (**c**) from 20 to 90 °C.

**Figure 2 foods-13-00546-f002:**
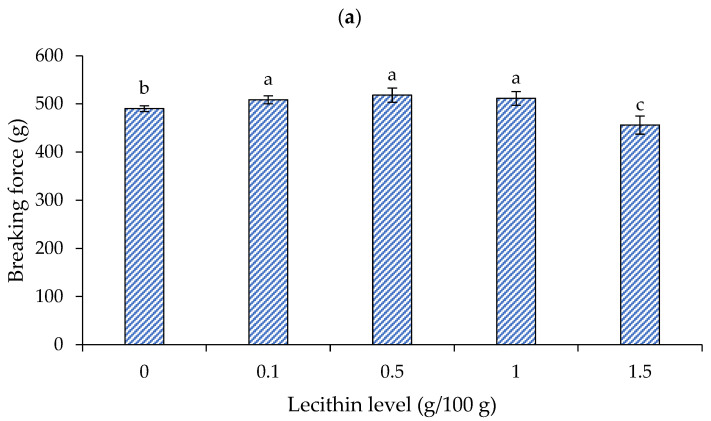
Breaking force (**a**) and deformation (**b**) of gels from single-washed mackerel surimi added with varied lecithin concentrations. The standard deviations from triplicate determinations are represented by the bars. Significant differences (*p* < 0.05) are denoted by different letters.

**Figure 3 foods-13-00546-f003:**
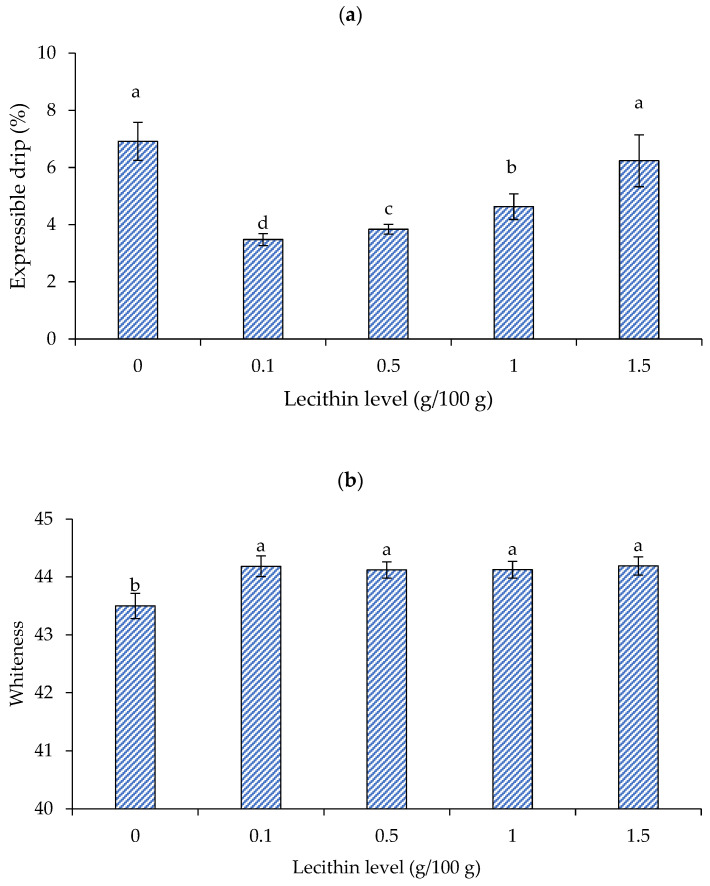
Expressible moisture (**a**) and whiteness (**b**) of gels from single-washed mackerel surimi added with varied lecithin concentrations. The standard deviations from triplicate determinations are represented by the bars. Significant differences (*p* < 0.05) are denoted by different letters.

**Figure 4 foods-13-00546-f004:**
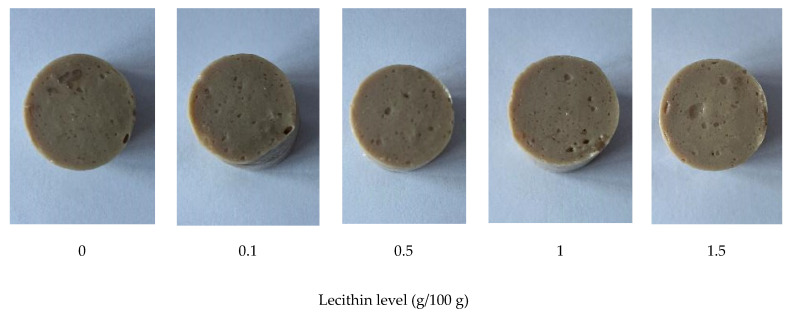
Appearance of gels from single-washed mackerel surimi added with varied lecithin concentrations.

**Figure 5 foods-13-00546-f005:**
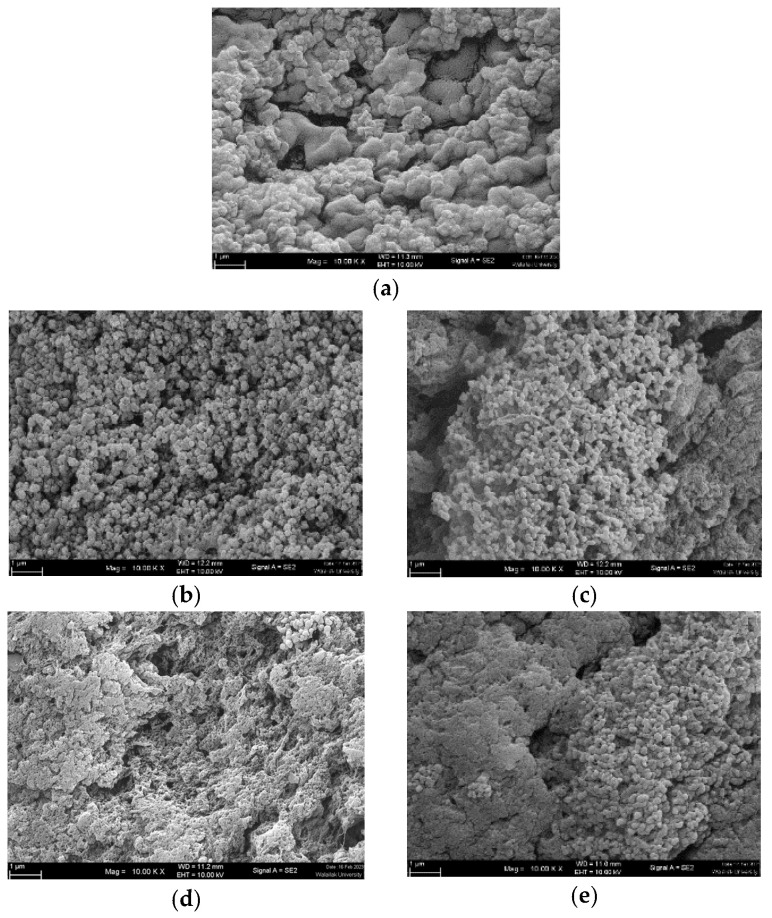
Electron microscopic images of gels from single-washed mackerel surimi added with varied lecithin concentrations. (**a**) = 0 g/100 g, (**b**) = 0.1 g/100 g, (**c**) = 0.5 g/100 g, (**d**) = 1 g/100 g, (**e**) = 1.5 g/100 g). Magnification: 10,000×, EHT: 5 kV.

**Figure 6 foods-13-00546-f006:**
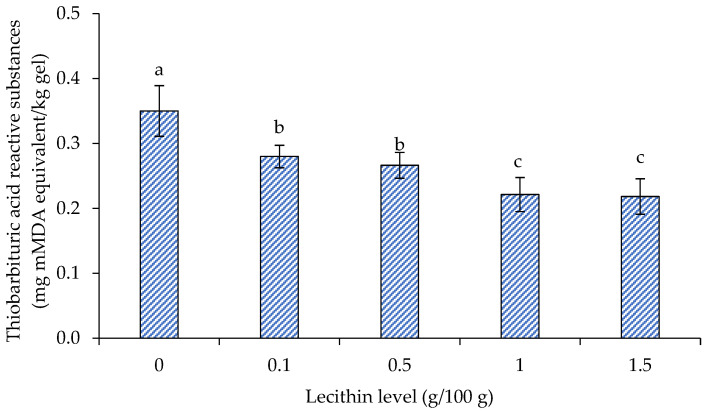
Thiobarbituric acid reactive substances (TBARS) of gels from single-washed mackerel surimi added with varied lecithin concentrations. The standard deviations from triplicate determinations are represented by the bars. Significant differences (*p* < 0.05) are denoted by different letters.

## Data Availability

The original contributions presented in the study are included in the article, further inquiries can be directed to the corresponding author.

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
