# Peer review of "Relatively Low Lecithin Inclusion Improved Gelling Characteristics and Oxidative Stability of Single-Washed Mackerel (Auxis thazard) Surimi"

_foods, 2024, doi:10.3390/foods13040546_

Round 1
Reviewer 1 Report
Comments and Suggestions for Authors
Well written and interesting manuscript even for those that are not directly related to the topic. However, there are still some things that need to be added and changed in order to improve quality of the manuscript and clarify some statements and explanations. Those suggestions are given as comments in PDF document. In addition to those comments abstract is in my opinion to long and loaded with unnecessary data. It needs to be shortened a bit and instead of numbers I suggest that authors give some trends (like in summary/conclusions). Minor English check by some Grammarly software will be welcomed too.

Minor English grammarly checking.
Author Response
Reviewer 1
Well written and interesting manuscript even for those that are not directly related to the topic. However, there are still some things that need to be added and changed in order to improve quality of the manuscript and clarify some statements and explanations. Those suggestions are given as comments in PDF document.
Ans: Thank you very much. All of the inquiries in the PDF document were addressed point by point. See below.
In addition to those comments abstract is in my opinion to long and loaded with unnecessary data. It needs to be shortened a bit and instead of numbers I suggest that authors give some trends (like in summary/conclusions).
Ans: I frequently encountered reviewers who recommended including the numerical values of the results. Thus, certain data has to be given. Nevertheless, as recommended several unnecessary elements were removed from the abstract.
Minor English check by some Grammarly software will be welcomed too.
Ans: English was double-checked using a paraphrase program, QuillBot, and proofread by Prof.Dr. Kalidas Shitty of North Dakota State University in the United States, one of the co-authors.
Comments in PDF document.
Ans: Originally, it was stated that “The rheological behavior, gel-forming ability, microstructure, and lipid oxidation of lecithin-added surimi varied significantly depending on lecithin content.” Thus, it is the gel- forming ability of lecithin-added surimi.
Ans: It was changed to “When compared to the control (no lecithin addition), lecithin at 0.1, 0.5, and 1 g/100 g improved breaking force of the gel (p < 0.05).”
Ans: Before being used, lecithin was dissolved in cold water. In this instance, we applied just a little amount of chilled water to help the lecithin dissolve. Next, we adjust the final moisture content of each treatment to 80% in order to maintain an equal final moisture content in the surimi, as stated in the text “Before applying lecithin, it was dissolved in a small amount of chilled water. The final moisture content in the surimi mixture for all treatments was adjusted to 80% (w/w).”
Ans: The concentration of lecithin in this study was based on the weight of surimi as stated in the text “Pieces of surimi were chopped for a five-minute period with 2.5% (w/w) NaCl and varying amounts of lecithin (0, 0.1, 0.5, 1, and 1.5 g/100 g of surimi).”
Ans: It was changed to “The final moisture content in the surimi mixture for all treatments was adjusted to 80% (w/w).”
Ans: The statements were added.
- “G" is typically used to measure the viscosity of the surimi gel and refers to the loss modulus [23].”
- “Thus, it appears that lecithin had a concentration-dependent effect on the viscoelastic characteristics of surimi, which was linked to the gel strength of surimi.”
Ans: It was changed to “Consequently, due to the rheological behavior, two-step heating (40 °C/30 min and 90 °C/20 min) can be used for the production of gels where all samples can gel to varying degrees.”
Ans: Yes, it is. So, we stated originally that “Then, G' decreased immediately, touching its lowest point at 50 °C for surimi with no or low lecithin content (0-0.5 g/100 g) and about 60 °C for surimi with 1-1.5 g/100 g, suggesting gel weakening.”
Ans: Thank you very much. The statement was added. “Additionally, surimi with low or no lecithin levels (0-0.5 g/100 g) had nearly identical G" values up to 65 °C.”
Ans: It was added. “…. and the gel containing lecithin at 1.5 g/100 g showed significantly decreased deformation (~20% decrease).”
Ans: We updated the figure to the same resolution as the marked 1 micrometer.

Reviewer 2 Report
Comments and Suggestions for Authors
The paper "Relatively Low Lecithin Inclusion Improved Gelling Character- 2 istics and Oxidative Stability of Single-Washed Mackerel 3 (Auxis thazard) Surimi" is an interesting topic, in its current form, the paper should only be considered for publication after Minor revision. In a subsequent submission, the authors should address the following:
1. What is the basis for setting the different lecithin concentrations at 0, 0.1, 0.5, 1, 15 and 1.5 g/100 g of surimi?
2. L110-L111“modern customer demand”, what kind of needs does that mean?
3. In Figure 1 (a), please use commas to separate thousands for numbers with five or more digits (not four digits) in the image, e.g., "10,000" should be "10,000".
4. The limitation of this study and the direction of future research can be increased in the conclusion part.
Author Response
Reviewer 2
The paper "Relatively Low Lecithin Inclusion Improved Gelling Character- 2 istics and Oxidative Stability of Single-Washed Mackerel 3 (Auxis thazard) Surimi" is an interesting topic, in its current form, the paper should only be considered for publication after Minor revision. In a subsequent submission, the authors should address the following:
- What is the basis for setting the different lecithin concentrations at 0, 0.1, 0.5, 1, 15 and 1.5 g/100 g of surimi?
Ans: We originally stated in Section 2.3 that “The lecithin concentration employed in this study was based on our prior evidence, which suggested using lecithin at 1 g/100 g for bigeye snapper surimi [18]. The type of surimi raw material and the washing procedure, however, may play a vital role in determining how much lecithin is integrated. As a result, for the single-washed mackerel surimi, lecithin concentrations at lower and higher than 1 g/100 g were used in this investigation.”
- L110-L111“modern customer demand”, what kind of needs does that mean?
Ans: It was changed to “The results collected can be used to advance health-oriented surimi products prepared from dark-fleshed fish, adhering to the growing demand for products with functional ingredients.”
- In Figure 1 (a), please use commas to separate thousands for numbers with five or more digits (not four digits) in the image, e.g., "10,000" should be "10,000".
Ans: The journal's format suggests the use of commas beginning with 10,000.
- The limitation of this study and the direction of future research can be increased in the conclusion part.
Ans: The statement was added. “However, sensory evaluation and storage stability can be further investigated to meet industrial requirements. In addition, the combined effect of lecithin and other functional food ingredients on the gel characteristics of mackerel surimi can be investigated in the future.”

Reviewer 3 Report
Comments and Suggestions for Authors
The authors reported the preparation of single-washed Mackereñ surimi with five different concentrations (0, 0.1, 0.5, 1.0 and 1.5 g/100g) of lecithin. The resulted surimi products were investigated under different techniques such as: Rheological properties, Gelling properties based on their breaking force and deformation as well as their expressible drip and whiteness. Also, SEM images were reported to show the appearance of gels.
The manuscript is coherent between the sections specially at the results and discussion section. There are some minor issues to attend:
- Page 1 L26-27. Rewrite the sentence
- Page 4 L158. Include a brief description of the methods. It includes: equipment, configuration, sample form and dimensions.
- Page 4 L161. Include a brief description of the methods. It includes: equipment, configuration, etc.
- Page 4 L164. Include a brief description of the methods. It includes: equipment, configuration, etc.
- Page 4 L168. What is the experimental design?
- Figures 2,3 and 6. Use the “a” letter for higher values.
- Page 12 L380-386. Delete discussion of results from conclusions section.
- Page 12 L399. Delete this line
Comments on the Quality of English Language
Minor editing of English language required
Author Response
Reviewer 3
The authors reported the preparation of single-washed Mackerel surimi with five different concentrations (0, 0.1, 0.5, 1.0 and 1.5 g/100g) of lecithin. The resulted surimi products were investigated under different techniques such as: Rheological properties, Gelling properties based on their breaking force and deformation as well as their expressible drip and whiteness. Also, SEM images were reported to show the appearance of gels.
The manuscript is coherent between the sections specially at the results and discussion section. There are some minor issues to attend:
- Page 1 L26-27. Rewrite the sentence
Ans: It was changed to “Adding lecithin to mackerel surimi improved its whiteness slightly regardless of concentration.”
- Page 4 L158. Include a brief description of the methods. It includes: equipment, configuration, sample form and dimensions.
Ans: Done.
By using the technique of Buamard and Benjakul [21], gel properties such as breaking force, deformation, whiteness, and expressible drip were examined. Breaking force and deformation of gel with 2.5-cm length were determined using a TA-XT2i texture analyzer (Stable Micro Systems, Godalming, Surrey, UK), equipped with a spherical plunger (5-mm diameter and speed of 60 mm/min). The expressible drip was calculated based on the percentage of the sample weight before and after being compressed by 5-kg standard weight for 2 min where a thin sliced sample (5-mm thickness) was placed between pieces of Whatman No. 1 filter paper. The color parameters including L*, a*, and b* values were recorded using a portable Hunterlab Miniscan/EX instrument (Hunter Assoc. Laboratory, VA, USA) and the whiteness was calculated by the following formula:
Whiteness = 100 - [(100 - L*)2 + a*2 + b*2]1/2 (1)
- Page 4 L161. Include a brief description of the methods. It includes: equipment, configuration, etc.
Ans: Done.
Surimi gel microstructure examination was carried out using the scanning electron microscope (SEM) (GeminiSEM, Carl Ziess Microscopy, Germany) [2]. Samples with a thickness of 2-3 mm were fixed with 2.5% (v/v) glutaraldehyde in 0.2 M phosphate buffer (pH 7.2) for 2 h. The samples were then rinsed for 1 h in distilled water before being dehydrated in ethanol with serial concentrations of 50, 70, 80, 90, and 100% (v/v). Dried samples were mounted on a bronze stub and sputter-coated with gold. The specimens were observed with an SEM at an acceleration voltage of 5 kV.
- Page 4 L164. Include a brief description of the methods. It includes: equipment, configuration, etc.
Ans: Done.
As a measure of lipid oxidation, thiobarbituric acid reactive substances (TBARS) of surimi gels were examined [22]. Ground sample (0.5 g) was homogenized with an IKAÒ homogenizer (Model T25 digital Ultra-TurraxÒ, Staufen, Germany) with 2.5 mL of a solution containing 0.375% (w/v) TBA, 15% (w/v) TCA, and 0.25 M HCl at 9500 rpm for 2 min in an ice bath. The mixture was heated in a boiling water bath (95-100 °C) for 10 min, cooled with running tap water, and then centrifuged (3600 ×g/ 25 °C/ 20 min). The absorbance of the supernatant was measured at 532 nm. A standard curve was prepared using 1,1,3,3-tetramethoxypropane at concentrations ranging from 0 to 10 mg/L. Malondialdehyde (MDA) equivalent in mg/kg of gel was reported as the TBARS content.
- Page 4 L168. What is the experimental design?
Ans: It was stated that “Completely randomized design (CRD) was applied for all examinations. Data are presented as mean ± standard deviation from triplications. The Statistical Package for the Social Sciences for Windows (SPSS Inc., Chicago, IL, USA) was used to process the data. Significant differences (p < 0.05) between samples were found using Duncan's multiple-range analysis.”
- Figures 2,3 and 6. Use the “a” letter for higher values.
Ans: Done.
- Page 12 L380-386. Delete discussion of results from conclusions section.
Ans: We would like to keep it. It was not the discussion. We would like to emphasize the effect of lecithin concentration on surimi gel characteristics in order to reach a conclusion about the optimal lecithin concentration in the final section. Thank you very much.
- Page 12 L399. Delete this line
Ans: Data Availability Statement is the format of the journal. So, we would like to keep it.
Comments on the Quality of English Language
Minor editing of English language required
Ans: English was double-checked using a paraphrase program, QuillBot, and proofread by Prof.Dr. Kalidas Shitty of North Dakota State University in the United States, one of the co-authors.
